# Ultrastructure of the nebenkern during spermatogenesis in the praying mantid *Hierodula membranacea*

**Maria Köckert**[1☉], **Chukwuebuka William Okafornta**[1☉], **Charlice Hill**[1], **Anne Ryndyk**[1], **Cynthia Striese**[1], **Thomas Müller-Reichert**[1], **Leocadia Paliulis**[2], **Gunar Fabig**[1]*

1 Experimental Center, Faculty of Medicine Carl Gustav Carus, Technische Universität Dresden, Dresden, Germany, 2 Biology Department, Bucknell University, Lewisburg, PA, United States of America

☉ These authors contributed equally to this work.
* gunar.fabig@tu-dresden.de

## Abstract

Spermatogenesis leads to the formation of functional sperm cells. Here we have applied high-pressure freezing in combination with transmission electron microscopy (TEM) to study the ultrastructure of sperm development in subadult males of the praying mantid *Hierodula membranacea*, a species in which spermatogenesis had not previously been studied. We show the ultrastructure of different stages of sperm development in this species. Thorough examination of TEM data and electron tomographic reconstructions revealed interesting structural features of the nebenkern, an organelle composed of fused mitochondria that has been studied in spermatids of other insect species. We have applied serial-section electron tomography of the nebenkern to demonstrate in three dimensions (3D) that this organelle in *H. membranacea* is composed of two interwoven mitochondrial derivatives, and that the mitochondrial derivatives are connected by a zipper-like structure at opposing positions. Our approach will enable further ultrastructural analyses of the nebenkern in other organisms.

## Introduction

The praying mantids (order Mantodea) have been used as a system for studying spermatocyte meiosis for over a century, with the first publication describing aspects of mantid male meiosis in 1897 [1]. Unique features of chromosome behavior during both meiosis I and meiosis II have contributed greatly to our understanding of chromosome condensation, the balance of forces in spindles, and the spindle checkpoint [2–9]. While many insights into spermatocyte meiosis have resulted from studies using praying mantids, there are few reports of other stages of spermatogenesis in mantids. The only survey of spermatogenesis in a praying mantid was performed by Williams [10], who studied this process in the mantid *Choeradodis rhombicollis*, showing drawings of different stages of spermatogenesis in fixed and stained specimens observed through a light microscope. More recently, electron microscopy of mantid sperm [11, 12], and male reproductive anatomy [13] have also been studied, but a thorough survey of

**Data Availability Statement:** All raw data (light microscopy and TEM) and reconstructed electron tomograms with segmentations are available from the Open Access Repository and Archive (OpARA)

of TU Dresden: https://opara.zih.tu-dresden.de/xmlui/handle/123456789/5878.

**Funding:** National Science Foundation (NSF, RUI-1715157 to L.P.) https://www.nsf.org/ Deutsche Forschungsgemeinschaft (DFG, MU1423/10-1 to T.M-R.) https://www.dfg.de/ The funders had no role in study design, data collection and analysis, decision to publish, or preparation of the manuscript.

**Competing interests:** The authors have declared that no competing interests exist.

spermatogenesis including ultrastructural features in a praying mantis has not been published so far. Such a detailed study on the structure of cells in the testes of a mantid by both light and transmission electron microscopy (TEM) has the potential to reveal a great deal about sperm development in general, providing an additional insect order for comparative work. Because the cells are large, can lie flat on a slide for light microscopy, and have large, clearly visible organelles under any type of microscopy, they allow for detection of more difficult to see cytological details, thus informing previous studies in other organisms.

Some of the earliest studies of spermatogenesis in insects revealed the existence of a relatively large, layered structure made up of mitochondria, which was called the nebenkern [14, 15]. The nebenkerns of several insect species (Hemiptera, Heteroptera) were characterized by light microscopy [15, 16] and later by electron microscopy [17, 18], showing that the nebenkern forms at the end of meiosis II through the fusion of mitochondria. The nebenkern is composed of two halves separated by a furrow, with interdigitating layered membranes and thin bridges connecting the two halves [17]. Further studies of spermatogenesis in *Drosophila melanogaster* revealed that this insect also forms a nebenkern during spermatogenesis that also displays a layered so-called "onion" structure, composed of two halves divided by a furrow [19, 20]. Examination of spermatogenesis in *D. melanogaster* allowed the merging of both light microscopical and ultrastructural data, and molecular genetics, showing that the initial fusion of the mitochondria to form the nebenkern requires intact microtubules, the dynamin-related protein Drp1, and components of the pink1/Parkin pathway [21]. Formation of a functional and visibly wild-type nebenkern also requires other proteins that mediate mitochondrial fusion, like Fzo (an orthologue of the human mitofusins, Mfn1 and Mfn2), and the testis-specific ATP synthase subunit d paralog knon [22–24]. Later in spermatogenesis, the two mitochondrial derivatives of the nebenkern unwrap and elongate along the axoneme, distributing mitochondria so they can play their important role in powering motion of the sperm tail [16].

Previous studies in insects have revealed interesting membrane features associated with spermatogenesis, variations in flagellar structure, and variations in the structure and fusion of mitochondria associated with the formation of the nebenkern [12, 17]. The aim of this study is to show new information about spermatogenesis and nebenkern structure through an initial characterization of cells in the testes of the praying mantid *Hierodula membranacea*. Using state-of-the-art sample preparation techniques in combination with three-dimensional (3D) imaging such as high-pressure freezing and serial-section electron tomography, we show a survey of the steps of spermatogenesis in this species. Our study reveals interesting features of cellular structures during spermatogenesis, including the structure of the chromosomes in prophase I and the distribution of organelles in spermatids. In particular, we show the existence of the nebenkern in *H. membranacea* spermatids, revealing that the nebenkern of this species has two zipper-like membrane structures at opposing positions.

## Materials and methods

### Handling of subadult mantids

Subadult males of *Hierodula membranacea* (Giant Asian Mantis) were obtained from InsectSales.com (Port Angeles, Washington, USA) and Mantids & More (Mühlheim am Main, Germany). The authors verified the vendor's identification according to previous publications [25, 26]. Mantids were fed house flies and crickets according to instructions from the vendors. Subadult males were selected for experimentation because that is the only stage of praying mantid development in which all stages of spermatogenesis can be found in the testes. To remove testes from subadult males, slits were made on both sides of the posterior of the abdomen, just anterior to the last abdominal segment. Testes were gently squeezed out of the slits,

removed with tweezers and then placed in either Voltalef oil (Atofina) for phase contrast light microscopy or phosphate buffer (0.1 M, pH 7.4) for preparation for electron microscopy.

## Light microscopy

Live-cell imaging of subadult male testes was performed as described in Lin et al. [27]. Briefly, spermatids at different stages of development were imaged across multiple focal planes using a Nikon Eclipse TS100 microscope (Nikon Instruments Inc., New York, USA) equipped with a 100x, 1.25 NA phase-contrast, oil immersion objective. Images were recorded using a View4K HD camera (Microscope Central) within the InFocus software (Microscope Central).

## Electron microscopy

**High-pressure freezing, freeze substitution and resin embedding.**   For ultrastructural preservation of the male gonad of *H. membranacea*, high-pressure freezing in combination with freeze substitution was applied. Cryo-immobilization of the gonads was achieved by cooling the sample down to liquid nitrogen temperature (-196˚C), while exposing a pressure of approx. 2000 bar [28]. In preparation for freezing, sample holders (type-A aluminum planchettes, Wohlwend, Switzerland) were pre-wetted with 1-hexadecene (Sigma). For each freezing run, the 200 µm indentation of a planchette was then filled with 0.1 M phosphate buffer containing 10% (w/v) polyvinylpyrrolidone (Sigma, MW 10,000 [28–30]), and pieces of the dissected mantid testes were transferred to the cavity. The type-A planchette was then closed with the flat side of 1-hexadecene-pre-wetted type-B planchette (Wohlwend, Switzerland). These 'sandwiches' were then immediately frozen using a high-pressure freezer (HPF Compact 03, Wohlwend, Switzerland) and stored in liquid nitrogen.

For subsequent freeze substitution, the cryo-immobilized samples were transferred under liquid nitrogen to a frozen 'cocktail', consisting of 0.1% (w/v) uranyl acetate (Polysciences, USA) and 1% (w/v) osmium tetroxide (EMS, USA) in anhydrous acetone (EMS, USA). The freeze substitution was performed with an automated freeze-substitution machine (AFS2, Leica, Austria). The samples stayed at -90˚C for one hour. The temperature was then raised to -30˚C with incremental steps of 5˚C per hour. The samples were kept at -30˚C for 5 hours. Afterwards, the temperature was raised again by 5˚C per hour to 0˚C, and the samples were kept for three additional hours until they were further processed.

**Thin-layer embedding, serial sectioning and post-staining.**   After freeze substitution, samples were washed three times in pure acetone and infiltrated with resin as previously published [31]. In brief, samples were gradually infiltrated with Epon/Araldite resin (one part resin: three parts acetone) for 1 hour; 1: 1 for 2 hours; 3: 1 for 2 hours, and 100% resin for 1 hour, then 100% resin overnight, then 100% resin for 1 hour and thin-layer embedded [31]. Resin-infiltrated samples were polymerized for three days at 60˚C.

Selected samples were re-mounted on dummy blocks for ultramicrotomy [31]. Serial thin (70 nm) sections for routine transmission electron microscopy and semi-thick (300 nm) sections for electron tomography were cut using an ultramicrotome (EM UC6, Leica Microsystems, Austria) equipped with a diamond knife (Diatome, Switzerland). Sections were collected on Formvar-coated copper slot grids and post-stained with 2% (w/v) uranyl acetate (Science Services, USA) in 70% methanol for 10 min, followed by 0.4% (w/v) lead citrate (Science Services, USA) in double-distilled water for 5 min. In addition, colloidal gold (20 nm diameter, BBI, UK) was attached to the semi-thick sections to serve as fiducial markers for the calculation of electron tomograms.

**TEM of thin sections and pre-screening of semi-thick sections.**   To inspect thin sections, grids were analyzed by using a transmission electron microscope (Morgagni, Thermo Fisher)

operated at 80 kV and equipped with a 2k x 2k CCD camera (Veleta, EMSIS). Thin sections were screened for developing and mature sperm cells. Semi-thick serial sections were also pre-screened using the same TEM to map regions of interest for subsequent 3D reconstruction by electron tomography.

**Electron tomography and 3D reconstruction.** Electron tomography was performed by using a transmission electron microscope (Tecnai F30, Thermo Fisher) operated at 300 kV and equipped with a 4k x 4k CMOS camera (OneView, Gatan). Using a dual-axis specimen holder (Type 2040, Fischione, USA), tilt series were recorded at a magnification of 4700x and a pixel size of 2.572 nm from -60˚ to +60˚ with 1˚ increments applying the SerialEM software package [32, 33]. For dual-tilt electron tomography, the grids were rotated for 90˚ in the XY-plane and the second tilt series was acquired using identical microscope settings [34]. Resulting tomographic A- and B-stacks of the same positions were reconstructed, combined and flattened using IMOD [35, 36].

Serial-sections of the nebenkern were stitched and combined [37, 38] by using both automatically segmented microtubules as initial landmarks [39, 40] and manually selected features applying the ZIB Amira (Zuse Institute Berlin, Germany) software package [41]. To automatically segment the membranes in the stitched serial tomograms of the nebenkern, image processing was performed within ZIB Amira that included the following computational operations: Morphological Laplacian (3D, 12 px, precision 'faster'), Gaussian Filter (3D, separable, SD 5x5x5 px, kernel size factor '3'), Grayscale Fill Holes (3D, neighborhood '18'), Adaptive Histogram Equalization (3D, contrast limit '3'), and Adaptive Thresholding (3D, window 200x200x30 px, threshold '75', criterion 'greater-or-equal', threshold mode 'additive'). The binary image stack output was then used to generate a 3D surface. The same image processing pipeline was applied also to the single-section tomograms to segment membranes and chromatin. The central elements of the polycomplexes and the microtubules of the basal body were manually segmented. The microtubules of the axoneme and all other microtubules were automatically segmented using the ZIB Amira software package [39, 40]. Movies were also rendered using this software.

## Results

### Ultrastructure of spermatocytes in *H. membranacea*

The testes of *H. membranacea* were observed to have a large number of follicles, and individual testicular follicles could be observed macroscopically. The combination of ultrarapid high-pressure freezing after the dissection, freeze substitution with 1% osmium tetroxide and 0.1% uranyl acetate yielded an excellent preservation of the ultrastructure of the testes tissue for electron microscopy.

In each cell identified as being in prophase I we observed a nucleus surrounded by mitochondria. Golgi and ER could often be observed at the periphery of the cells (Fig 1A and 1B). The nucleolus was often very prominently visible as dark stained structure inside of the nucleus (Fig 1B and 1C). In prophase I, paired homologous chromosomes were observed with the central element of the synaptonemal complex (SC) visible as a thin line in between the aligned chromatin (Fig 1C). We also observed multiple synaptonemal complexes stacked upon each other, structures called 'polycomplexes' (Fig 1C; [42]). Electron tomography of a prophase nucleus showed that within one nucleus there exist regions with a single central element of the SCs and others with two or three (Fig 1DI, 1DII; S1 Movie). We measured the diameter of the SC to be 120–130 nm and found that the periodicity in between the stacked central elements within the polycomplexes was 110–120 nm. In further developed spermatocytes, we observed cells in prometaphase II in which the nuclear envelope has been broken down and

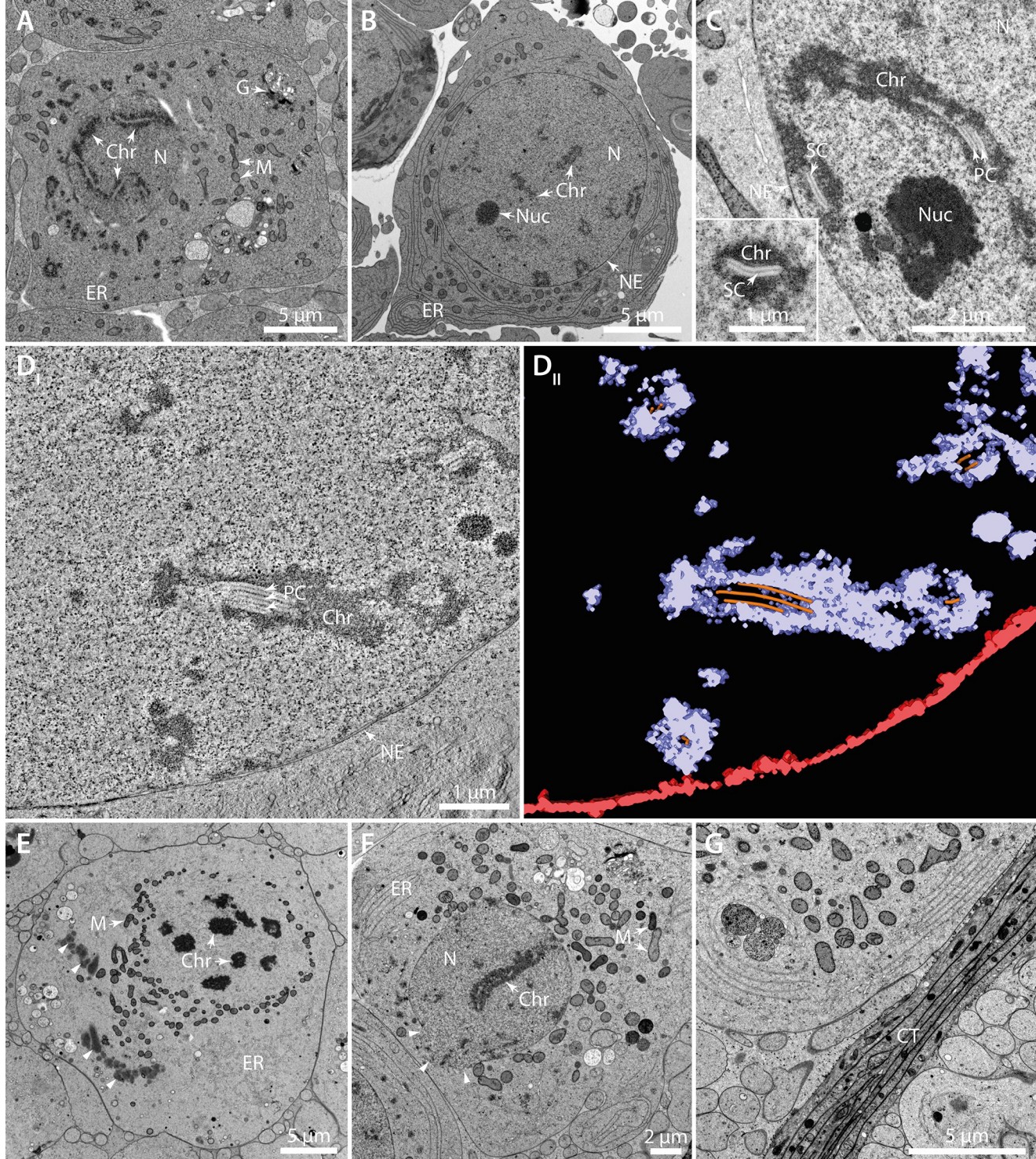

**Fig 1. Ultrastructure of early spermatocyte development in *H. membranacea*.** (**A**) TEM image of a spermatocyte in prophase of meiosis I. The nucleus (N) exhibits condensed chromatin (Chr) with synaptonemal complexes. Other organelles like the Golgi apparatus (G), the endoplasmic reticulum (ER) and mitochondria (M) are positioned in the vicinity of the nucleus. (**B**) Spermatocyte in prophase I. The nucleus (N) with a visible nuclear envelope (NE), a nucleolus (Nuc) and paired homologous chromosomes (Chr) are shown. The endoplasmic reticulum (ER) is positioned adjacent to the nucleus. (**C**) High-magnification view of a nucleus with condensed chromatin and a nucleolus. The central elements of synaptonemal complexes appear as single, double or triple complexes, a phenomenon known as polycomplexes (PC). The inset shows a synaptonemal complex at higher magnification. (**D$_I$**) Tomographic slice of a 3D reconstruction of a nucleus in prophase I. In this semi-thick section, the chromatin is positioned around multiple synaptonemal complexes and

polycomplexes. (**D_II**) 3D model of the nuclear envelope (red), the chromatin (purple) and central elements of the polycomplexes (orange). Same cell as shown in (**D_I**). (**E**) TEM image of a cell at prometaphase I. The chromosomes are condensed and the nuclear envelope is broken down. Mitochondria surround the chromosomes. The ER and electron-dense membrane-free aggregates (arrowheads) are visible in the cytoplasm. (**F**) TEM image of a spermatocyte presumably in meiosis II. The nuclear envelope appears open (arrow heads). (**G**) TEM of connective tissue (CT) separating the individual testicular follicles. The images are not shown according to the correct sequence of the developmental stages.

chromosomes showing a high level of condensation (Fig 1E). The mitochondria were arranged asymmetrically around the chromosomes in these cells. We also noticed ER or nuclear envelope remnants at the cell periphery and membrane-less protein aggregates of unknown origin or function (Fig 1E, arrowheads). It appeared evident that the accumulation of mitochondria begins to be asymmetric within later meiotic stages (Fig 1F). In some instances, we observed that the nuclear envelope is incomplete or not completely reformed (Fig 1F, arrowheads). Between the individual testicular follicles, we observed a connective tissue of multiple cell layers separating them from one another (Fig 1G).

## Formation of the nebenkern

When investigating the TEM images of the testes sections, we noted the existence of the nebenkern in spermatids, which is derived from fused mitochondria. This organelle later matures while wrapping around the axoneme, thereby providing the necessary energy for the movement of the sperm tail in mature sperm cells [20, 43]. Investigation of spermatids by phase contrast light microscopy revealed a dark, crescent shaped structure next to the nucleus (Fig 2A). In later stages, this structure appeared more rounded and exhibited internal features (Fig 2B), which resembles the structures we observed by TEM. In subsequent stages, it appeared less dense as observed with phase contrast light microscopy and internal membrane structures were more apparent (Fig 2C). In some instances, we observed a dark structure at the nucleus that the pointed end of the nebenkern seemed to be pointing to. We suggest that this is a centriole adjunct (Fig 2C and 2D), which tethers the basal body to the nucleus and provides structural support for the outgrowing axoneme [44]. We also found a long dark structure spanning the distance from the nucleus to the cell membrane, which was surrounded by the very elongated nebenkern (Fig 2D). We speculate that this structure is the axoneme, and the cell is about to grow the sperm's tail.

By TEM we observed that in telophase II the mitochondria begin to cluster together actively at one side of the nucleus (Fig 2E). This clustering becomes very prominent with progressing cell maturation (Fig 2F and 2G). We believe that the mitochondria start to fuse together at the time of dense clustering. Interestingly, in these cells we always observed an open nuclear envelope. Presumably, the nucleus shrinks in volume at this time and that this opening then serves to release the nucleoplasm (Fig 2F and 2G, arrow heads). In further developed cells, the mitochondria have fused largely to form the nebenkern, a complex three-dimensional (3D) arrangement of membranes that is always close to the nucleus to which the basal body is anchored. This attachment is achieved by a granular, amorphous structure described as the centriole adjunct (Fig 2H) [44].

From the basal body we observed the outgrowing axoneme in the direction of the nebenkern. In some instances, we also saw annulate lamellae in the male germ cells, an ordered structure of nuclear pore complexes within the endoplasmic reticulum and hence close to the nucleus or nebenkern (Fig 2I). Although the luminal volume of the nebenkern increases with more fusion events happening, there are still cristae visible inside within in the lumen itself as well as adjacent to the membrane (Fig 2I, arrow heads). In further developing cells the nebenkern changed its shape and became a multilayered, very elongated membrane organelle, whose

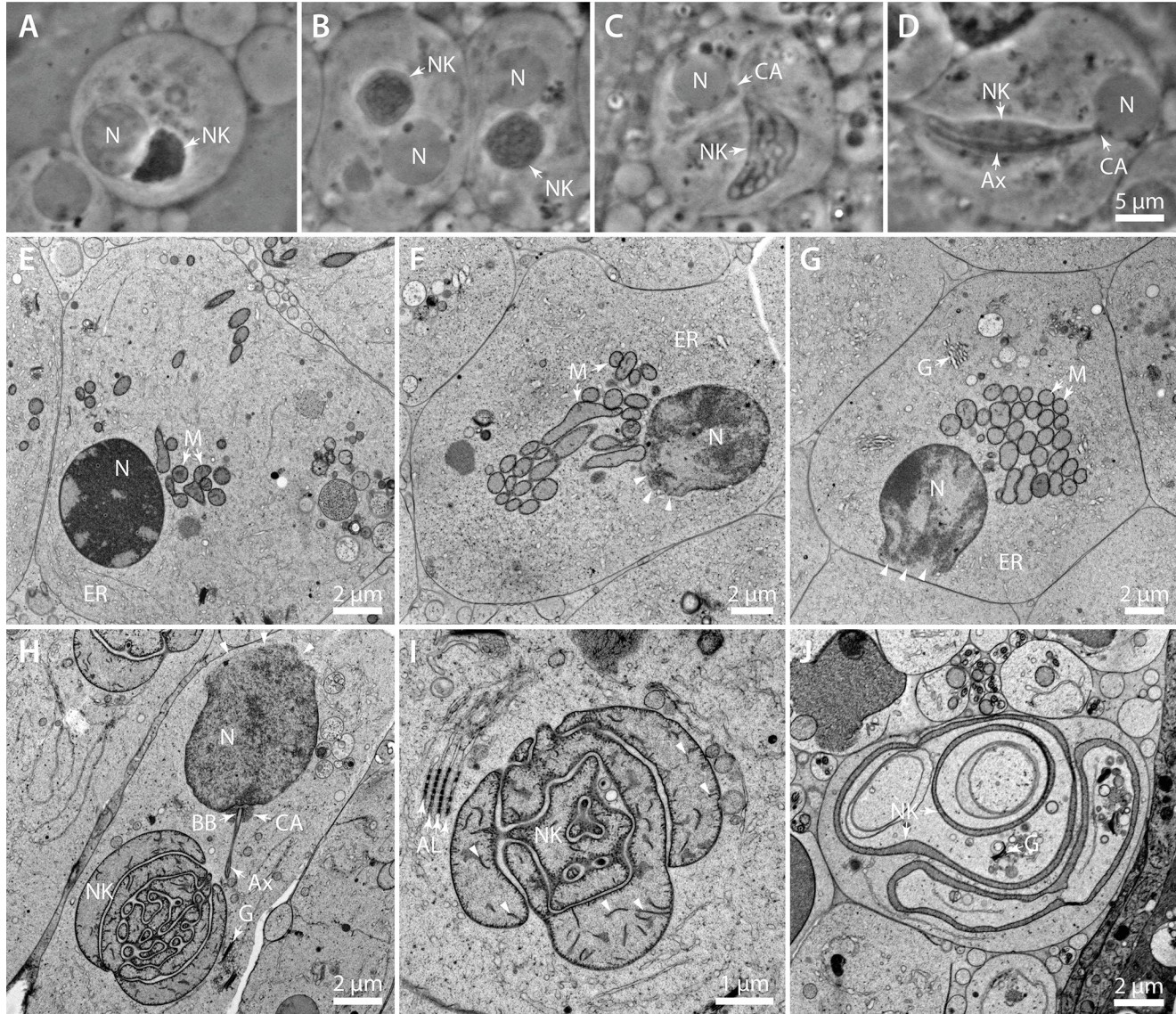

**Fig 2. Fine structure of differentiating sperm cells.** (**A**) Phase contrast image of a spermatid in an early stage of sperm maturation. The nucleus (N) and an irregular shaped nebenkern (NK) are shown. (**B**) Phase contrast image of a maturing spermatid. The nebenkern shows internal structures that cannot be resolved by phase contrast light microscopy. (**C**) Second example of a matured spermatocyte with an elongated, crescent-shaped nebenkern. Adjacent to the nucleus, a dark structure is visible that most likely represents a centriolar adjunct (CA) connecting the basal body to the nuclear envelope. (**D**) Spermatocyte with an axoneme (Ax) connected by the centriole adjunct material to the nucleus. The nebenkern appears elongated and spanning the whole volume of the cell. Scale bar for (A-D), 5 μm. (**E**) TEM image of a cell in late telophase II. The chromatin is still densely packed inside the nucleus, which is surrounded by endoplasmic reticulum (ER). Mitochondria (M) start to aggregate at a position close to the nucleus. (**F-G**) Sperm cell with clustered mitochondria, ER and Golgi (G). The nucleus (N) shows a fenestrated nuclear envelope (arrow heads). This fenestration is likely to be involved in a shrinkage and compaction of the nucleoplasm. (**H**) Nebenkern and nucleus with an attached basal body (BB) anchored through the centriolar adjunct structure (CA). The developing axoneme and the Golgi apparatus are visible in the vicinity of the nebenkern. The nuclear envelope appears fenestrated (arrowheads) at the opposite side of the nebenkern. (**I**) Maturing of the nebenkern into a complex multi-membrane structure. The cristae of the former mitochondria are still visible within the lumen of the nebenkern (arrowheads). Annulate lamellae (AL) are also visible presumably on an ER-derived structure next to it. (**J**) Nebenkern after maturation. In later sperm development, the nebenkern unfolds and assembles in the cytoplasm around the axoneme. In between this structure there are still cytoplasm and organelles like Golgi (G) visible.

shape can be hardly described by analyzing single thin TEM sections (Fig 2J). We noticed that in between the multiple membrane layers there are cellular organelles like Golgi and, therefore, the structure seems to fold around through the cytoplasm. In our understanding, the TEM

images shown in Fig 2H and 2I correspond to the light microscopic image shown in Fig 2B, and the TEM image in Fig 2J corresponds to Fig 2C and 2D.

## Three-dimensional reconstruction of the matured nebenkern

Previous work showed that the nebenkern consists of two independent but interlocked entities [20] as observed in the harlequin cabbage bug *Murgantia histrionica* [17]. Therefore, we sought to apply electron tomography to analyze the 3D architecture of the nebenkern in *H. membranacea*. We acquired electron tomograms from seven serial semi-thick (300 nm) sections (Fig 3AI–3IV). By stitching these serial tomograms, we were able to cover about one fourth of the volume of a nebenkern (Fig 3BI–3IV). After segmentation of the membranes in the tomographic sections, we then rendered the surfaces to obtain a 3D model of the nebenkern (Fig 3C; S2 Movie). We then displayed the inner volume of the nebenkern by clipping it along either the x- (Fig 3D; S3 Movie) or the y-axis (Fig 3E; S4 Movie) or both (Fig 3F). We observed that within the lumen of the nebenkern there were numerous membrane invaginations resembling the cristae of mitochondria. By further analyzing the 3D model we observed that the cytoplasm that appeared trapped within the nebenkern volume at a first glance was in fact not separated from the cytoplasm of the whole cell as it was not entirely enclosed by membranes.

In addition, we also observed two zipper-like membrane structures positioned directly opposite one another in our nebenkern reconstruction (Fig 3G–3J; S5 Movie). This zipper-like structure was not apparent when looking at single thin sections, emphasizing again the importance of 3D imaging for a spatial analysis of organelle organization. The membranes seemed to be connected in an alternating fashion (Fig 3G and 3I, arrow heads). We also observed an accumulation of microtubules at this region indicating a possible involvement of the cytoskeleton in this process (Fig 3H and 3J; S5 Movie).

Most importantly, however, our analysis revealed that there are two membrane entities that are inter-connected but both membrane systems never touched each other in the volume that has been investigated (Fig 3K–3N; S6 Movie). The 'zipper-like' structure at the outside is the region where both halves of the nebenkern are interwoven.

## Sperm maturation

In testicular follicles containing more mature sperm we observed axonemes next to slightly more electron-dense membrane structures showing membrane extrusions within their lumen which is presumably the matured nebenkern (Fig 4A and 4B). These structures were often obvious as round objects connected by a thin tube-like connection. In some instances, we found multiple axonemes next to each other (Fig 4A, arrow heads). In the investigated regions we also observed Golgi and thin membrane systems, which might be ER or its remnants. In some instances, we found that the nuclear envelope that is in contact with the centriole adjunct is densely stained thus appearing thickened (Fig 4C, arrowheads). The anchor that tethers the basal body to the nuclear envelope consisted of small darkly stained dots that resemble ribosomes in TEM images (Fig 4C). This motivated us to investigate this structure in detail using electron tomography. Within the reconstructed tomogram we saw that the nuclear envelope was in fact not thickened but appeared clearly bi-layered, with a darker staining at the outside facing the centriole adjunct (S7 Movie). These darkly stained structures within the centriole adjunct were 15–20 nm in diameter and appeared to be interconnected in a "beads-on-a-string" fashion (S7 Movie).

Tomographic reconstruction also revealed that the axoneme grows in the direction of the nebenkern which forms a pouch presumably resembling an opening or the bipartite split of the two nebenkern derivatives that engulfs the axoneme (Fig 4DI–4DIII; S8 Movie). In an area

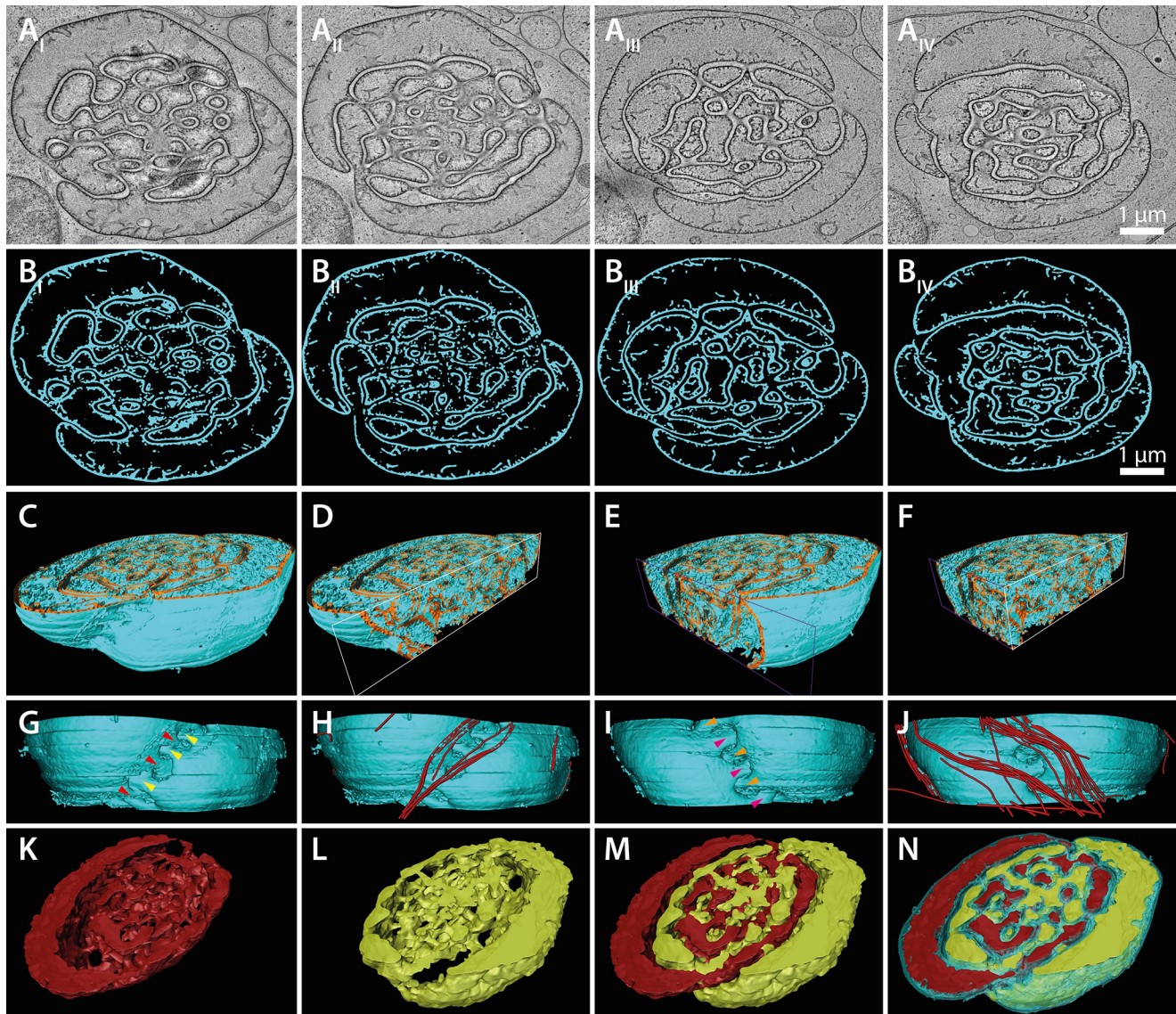

**Fig 3. Partial 3D reconstruction of the nebenkern covering about 2.1 μm in the z-dimension.** ($A_I$-$A_{IV}$) Tomographic slices through a nebenkern. ($B_I$-$B_{IV}$) Segmentation of membranes (cyan) in the tomographic slices as shown in (A). (**C-F**) Side views of the 3D model of the nebenkern with two clipping planes (x-axis white, y-axis purple) perpendicular to each other, illustrating the interlocking of the internal membranes of the nebenkern. The membrane is shown in cyan and then membrane lumen of the nebenkern is shown in orange. (**G**) Front view of the 3D model of the nebenkern illustrating a zipper-like membrane structure. The arrow heads (differently colored) illustrate the direction of the membrane extrusions leading into the nebenkern from opposite positions. (**H**) Identical view showing microtubules (red) in close proximity of the membrane structure. (**I-J**) Back view on the nebenkern model with respective microtubule model. (**K**) Segmentation of the interconnected inner lumen of one out of two mitochondria-derived nebenkern constituents (shown in red). (**L**) Segmentation of the other mitochondria-derived nebenkern constituents (shown in yellow). (**M**) 3D model of both nebenkern constituents illustrating the interlocking of both membrane systems. (**N**) 3D model of the nebenkern with the membrane (cyan) and both segmented inner lumina of the constituents (red and yellow).

with more mature sperm we acquired another tomogram in order to illustrate that the nebenkern next to the axoneme consists indeed of round structures that are connected by a tube-like membrane structure (Fig 4E; S9 Movie). They were always in close proximity to the axoneme and exhibited another set of microtubules in their cytoplasm that were clearly different from axonemal microtubules, as well as Golgi stacks (Fig 4E; S10 Movie).

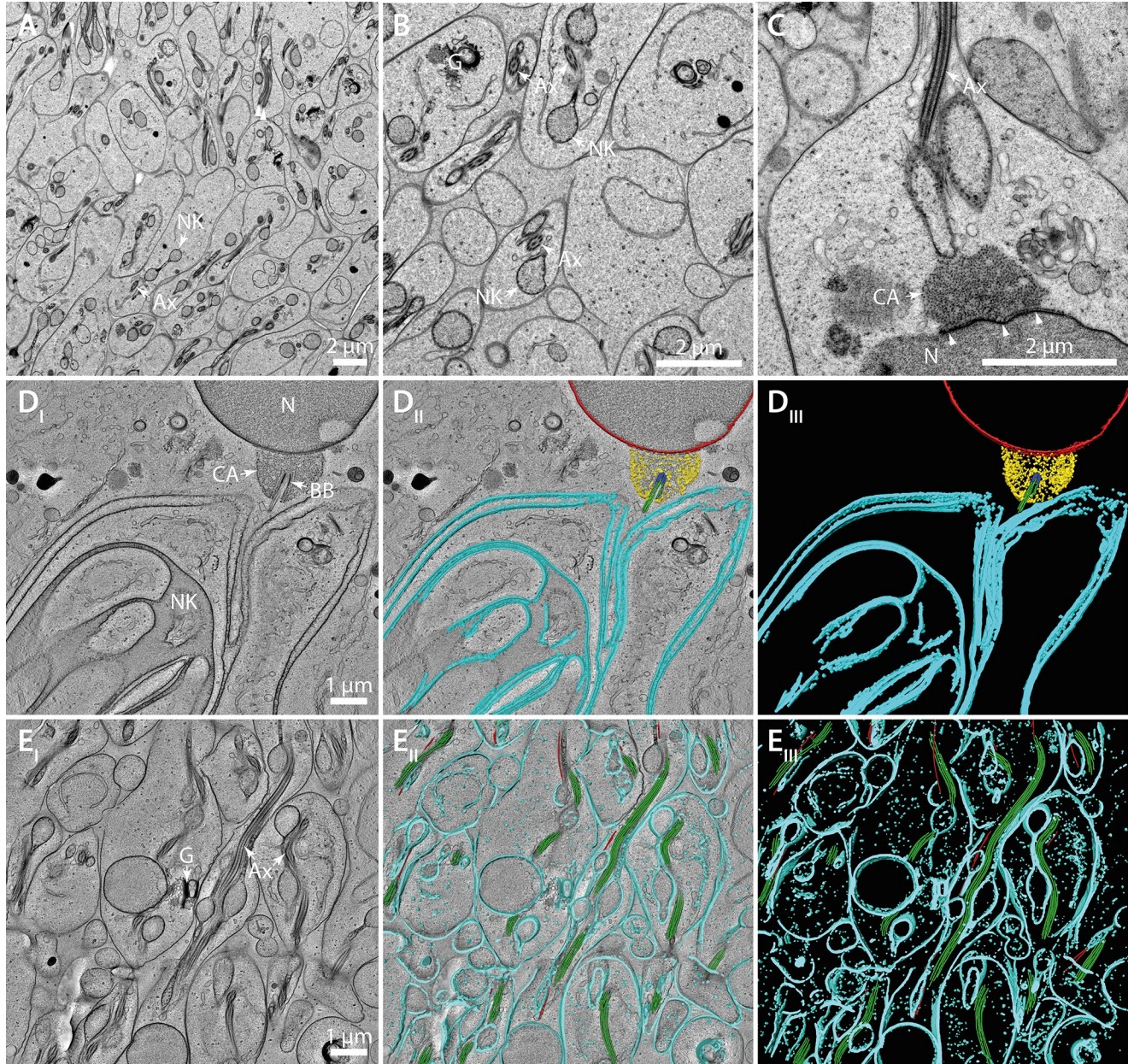

**Fig 4. Ultrastructure of spermiogenesis in *H. membranacea*.** (**A**) Thin-section-TEM image of a testes region. Multiple developing sperm cells are visible with axonemes (Ax) and nebenkern derivatives (NK). (**B**) Detailed view of a region with multiple Golgi complexes (G), axonemes (Ax) and nebenkern derivatives (NK) visible. (**C**) High-magnification view of a sperm cell illustrating the granular character of the centriolar adjunct (CA) next to the nucleus (N). An electron-dense structure is indicated (arrow heads). An outgrowing axoneme (Ax) is visible at an opposing position. ($D_I$-$D_{III}$) Electron tomogram and segmentation of the nucleus with an attached basal body (BB). The centriolar adjunct surrounds the basal body with the outgrowing axonemal microtubules. A developing nebenkern is visible next to the axoneme. The nuclear envelope is shown in red, the nebenkern membrane in cyan, the centriole adjunct in yellow, the basal body in blue and outgrowing microtubules in green. ($E_I$-$E_{III}$) Electron tomogram and segmentation of a region containing developing sperm tails. The Golgi complex and many axonemes are shown in cross- and longitudinal sections. In the 3D models, membranes are illustrated in cyan, axonemal microtubules in green and other microtubules in red.

## Discussion

By applying high-pressure freezing in combination with 3D reconstruction by serial-section electron tomography, we report on the ultrastructure of the nebenkern and other cytological

features during spermatogenesis in the praying mantid, *Hierodula membranacea*. In previous studies, the ultrastructure of male germ cells of other insect species was investigated in chemically fixed samples [45]. This classical method of fixation suffers from a slow rate of diffusion of the fixative into cells and tissues [45]. Depending on the sample and the occurrence of cuticles and other mechanical barriers, the diffusion rate of the fixative is less than one millimeter per hour and, therefore, fixation by diffusion can take up to minutes or several hours [46]. To avoid any preparation artifacts, we performed ultrarapid cryo-immobilization followed by freeze substitution [28]. Applying this approach, we achieved an excellent ultrastructural preservation.

Visualizing spermatids in a range of stages of maturation, the ultrastructure of spermatogenesis in *H. membranacea* at the first glance is similar to that observed in other insects [12]. Unexpectedly, however, we observed multiple synaptonemal complexes in stacks in areas within a single chromosome (Fig 1C), a phenomenon previously described as 'polycomplexes' [42]. We noted the existence of polycomplexes in prophase I. This structure has been documented as well in other eukaryotes (e.g. *D. melanogaster*, *S. cerevisia*e and *C. elegans*) [42].

An additional interesting observation is the plasticity and complexity of the nebenkern formation in this species. Until the end of meiosis II, multiple individual mitochondria are visible in the vicinity of the nucleus. In telophase II, there is an apparent accumulation of the mitochondria close to the nucleus. Although we do not have data on the dynamics of this process, we speculate that this mitochondrial accumulation is achieved through an association with microtubules. Support for this hypothesis might be provided by a detailed spatial analysis of microtubules in the vicinity of mitochondria. By addition of nocodazole to isolated spermatocytes, one could also perturb the cytoskeleton to test whether a disruption of microtubule dynamics might prevent the formation of the nebenkern.

A puzzling aspect of nebenkern formation is how the mitochondria fuse to form such an enormous organelle. Mitochondrial fusion is known to be mediated by the GTPases mitofusin-1 and mitofusin-2 (*Mfn1*, *Mfn2*) which are located in the outer mitochondrial membrane [23, 47]. In *D. melanogaster* it has been shown that fuzzy onions (fzo) mutants show defects in nebenkern formation and lead to sterility [22]. The question arises, do processes that are involved in the nebenkern formation in *H. membranacea* also rely on this mitochondrial fusion machinery? To test this, either RNAi and/or CRISPR/Cas technology would need to be developed in this species to inhibit mitofusin activity. The maturation of the nebenkern is also a fascinating cell biological topic. After a labyrinth-like state of the nebenkern is formed (Figs 2H, 2I and 3), this organelle completely flattens out to adopt a layered structure composed of thin sheets (Fig 2J), culminating in a tube formation (Fig 4A, 4B and 4E). The 3D organization and the molecular mechanisms of this morphological process also remain to be unraveled.

By segmenting the inner lumen of the nebenkern and grouping of connected parts we showed that the nebenkern at this stage is indeed composed of two parts (Fig 3K–3N). The complex nature of the interlocked segments and the fact that both parts did not seem to touch each other raises the question of how the two parts are built in the first place and how the partitioning then is achieved. Further structural studies of earlier and later nebenkern stages are clearly needed to better understand the morphological changes that take place during partitioning and subsequent flattening of this organelle.

Interestingly, we detected a zipper-like membrane arrangement on two opposing sides of the nebenkern (Fig 3G–3J; S5 Movie). This membrane shape was not obvious when looking at the segmented, stitched z-planes (Fig 3B). Only rendering of the membrane made it possible to analyze the 3D organization. In *Drosophila melanogaster* and *Murgantia histrionica*, the data suggest that the nebenkern splits into two derivatives, finally wrapping around the axoneme in the sperm tail [17, 20, 43]. If this holds true also for *H. membranacea*, it would explain the

observed two-fold symmetry of this structure and the observed 'zipper-like' structure could be a result of this process. The presence of microtubules on both 'zipper-like' membrane structures could indicate the involvement of the microtubule cytoskeleton and motor proteins in the separation or movement of this structure in the cell.

Our 3D electron microscopic analysis revealed that the nebenkern in *Hierodula membranacea* is indeed composed of two interwoven segments that are connected by a 'zipper-like' structure at opposing positions. Our ultrastructural approach could be applied to an analysis of spermatogenesis also in other insect species with similar structures to investigate if these features of nebenkern formation are evolutionarily conserved.

In conclusion, we show the first detailed description of the ultrastructure of cells in spermatogenesis in the praying mantid *Hierodula membranacea*, and the first detailed description by EM of mantid spermatogenesis. This work revealed that *H. membranacea* spermatids have a nebenkern that is composed of two intertwined segments that later separate into two mitochondrial derivatives and associate with the developing axoneme, as has been observed in other insects. Our use of novel techniques and 3D reconstructions reveal that the two intertwined segments of the nebenkern are joined by a zipper-like structure on the outside of the nebenkern.

## Supporting information

**S1 Movie. Electron tomographic reconstruction of a single semi-thick (300 nm) section of a nucleus in prophase I.** The nuclear envelope (red), the chromatin (purple), and the central elements (orange) of synaptonemal complexes and polycomplexes are shown. This movie corresponds to Fig 1D.
(MP4)

**S2 Movie. Movie of seven registered and stitched electron tomographic reconstructions of the nebenkern covering about 2.1 μm in the z-dimension.** The membrane of the nebenkern is shown in cyan, the membrane lumen in orange. This movie corresponds to Fig 3A–3C.
(MP4)

**S3 Movie. Stitched and segmented nebenkern with a consecutive clipping along the x-axis.** This 3D movie illustrates the structural complexity of the nebenkern. The membrane of the nebenkern is shown in cyan, the membrane lumen in orange. The clipping plane is indicated by a white rectangle. This movie corresponds to Fig 3D.
(MP4)

**S4 Movie. Stitched and segmented nebenkern with a consecutive clipping along the y-axis.** This 3D movie illustrates the structural complexity of the nebenkern. The membrane of the nebenkern is shown in cyan, the membrane lumen in orange. The clipping plane is indicated by a purple rectangle. This movie corresponds to Fig 3E.
(MP4)

**S5 Movie. Side view of the stitched and segmented nebenkern, illustrating a zipper-like membrane structure at both sides of the nebenkern.** The membrane of the nebenkern is shown in cyan. Microtubules (red) can be observed adjacent to the nebenkern membrane. This movie corresponds to Fig 3G–3J.
(MP4)

**S6 Movie. Segmentation of the interconnected inner lumen of the two mitochondria-derived nebenkern constituents (shown in red and yellow).** The membrane is shown in

cyan. This movie corresponds to Fig 3K–3N.
(MP4)

**S7 Movie. Electron tomographic reconstruction of a single semi-thick (300 nm) section of a nucleus with an attached basal body and the centriole adjunct.** This movie corresponds to Fig 4D and S8 Movie.
(MP4)

**S8 Movie. Electron tomographic reconstruction of a single semi-thick (300 nm) section of a nucleus with an attached basal body.** The centriole adjunct (yellow) surrounds the basal body (blue) with outgrowing axonemal microtubules (green). The nuclear envelope is shown in red, the developing nebenkern in cyan. This movie corresponds to Fig 4D and S7 Movie.
(MP4)

**S9 Movie. Electron tomographic reconstruction of a single semi-thick (300 nm) section of a region containing the developing axonemes and multiple nebenkern structures.** Axonemes in various orientations can be seen. Membranes are shown in cyan, axonemal microtubules in green and other microtubules in red. This movie corresponds to Fig 4E and S10 Movie.
(MP4)

**S10 Movie. Detailed view of an electron tomographic reconstruction of a single semi-thick (300 nm) section of a region showing the developing axonemes and multiple nebenkern structures.** This movie corresponds to Fig 4E and S9 Movie.
(MP4)

## Acknowledgments

We would like to thank the members of the Core Facility Cellular Imaging (CFCI, Faculty of Medicine Carl Gustav Carus, TU Dresden) for help with light microscopy and the EM facility at MPI-CBG, Dresden for technical assistance with electron tomography. We would also thank various members of the slack channel "EMofCellsTissuesOrganisms" for help with identifying structures. We are also grateful to Dr. Daniel Baum (Zuse Institut Berlin) for support in using ZIB Amira.

## Author Contributions

**Conceptualization:** Thomas Müller-Reichert, Leocadia Paliulis, Gunar Fabig.

**Data curation:** Maria Köckert, Chukwuebuka William Okafornta, Charlice Hill, Anne Ryndyk, Gunar Fabig.

**Formal analysis:** Maria Köckert, Chukwuebuka William Okafornta, Leocadia Paliulis, Gunar Fabig.

**Funding acquisition:** Thomas Müller-Reichert, Leocadia Paliulis.

**Investigation:** Maria Köckert, Chukwuebuka William Okafornta, Charlice Hill, Anne Ryndyk, Cynthia Striese, Thomas Müller-Reichert, Leocadia Paliulis, Gunar Fabig.

**Methodology:** Thomas Müller-Reichert, Leocadia Paliulis, Gunar Fabig.

**Supervision:** Thomas Müller-Reichert, Leocadia Paliulis, Gunar Fabig.

**Visualization:** Maria Köckert, Chukwuebuka William Okafornta, Charlice Hill, Anne Ryndyk, Thomas Müller-Reichert, Leocadia Paliulis, Gunar Fabig.

**Writing – original draft:** Leocadia Paliulis, Gunar Fabig.

**Writing – review & editing:** Maria Köckert, Chukwuebuka William Okafornta, Charlice Hill, Anne Ryndyk, Thomas Müller-Reichert, Leocadia Paliulis, Gunar Fabig.

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
