## [Decision Letter · Decision Letter 0]

17 May 2023

PONE-D-23-11181Ultrastructure of the nebenkern during spermatogenesis in the praying mantid *Hierodula membranacea*PLOS ONE

Dear Dr. Fabig,

Thank you for submitting your manuscript to PLOS ONE. After careful consideration, we feel that it has merit but does not fully meet PLOS ONE’s publication criteria as it currently stands. Therefore, we invite you to submit a revised version of the manuscript that addresses the points raised during the review process.

We look forward to receiving your revised manuscript.

Kind regards,

Wan-Xi Yang, Ph.D.

Academic Editor

PLOS ONE

Journal Requirements:

   "We would like to thank the members of the Core Facility Cellular Imaging (CFCI, Faculty of Medicine Carl Gustav Carus, TU Dresden) for help with light microscopy and the EM facility at MPI-CBG, Dresden for technical assistance with electron tomography. We would also thank various members of the slack channel “EMofCellsTissuesOrganisms” for help with identifying structures. We are also grateful to Dr. Daniel Baum (Zuse Institut Berlin) for support in using ZIB Amira. This work was supported by grants from the National Science Foundation (NSF, RUI1715157 to L.P.) and the Deutsche Forschungsgemeinschaft (DFG, MU1423/10-1 to T.M-R.)."

   "National Science Foundation (NSF, RUI-1715157 to L.P.)

https://www.nsf.org/

Deutsche Forschungsgemeinschaft (DFG, MU1423/10-1 to T.M-R.)

https://www.dfg.de/

Additional Editor Comments:

The current version of this manuscript cannot be considered for publication in PLOS One.  It needs a major revision. The following concerns should be addressed:

1. The abstract is very simple, the main conclusion is not included in the manuscript.

2. Improve the quality of some of the figures, refer to the reviewer's comments.

3. What is the main function of “nebenkern”?

4. Novelty of this study should be pointed in the "Conclusion" part.

All the two reviewers' concerns must be answered if you agree to revise your manuscript. If you decide to revise this manuscript, a list of changes should be uploaded along with the revision.

Reviewers' comments:

Reviewer's Responses to Questions

**Comments to the Author**

1. Is the manuscript technically sound, and do the data support the conclusions?

Reviewer #1: Yes

Reviewer #2: Yes

2. Has the statistical analysis been performed appropriately and rigorously? 

Reviewer #1: N/A

Reviewer #2: Yes

3. Have the authors made all data underlying the findings in their manuscript fully available?

Reviewer #1: Yes

Reviewer #2: Yes

4. Is the manuscript presented in an intelligible fashion and written in standard English?

Reviewer #1: Yes

Reviewer #2: Yes

5. Review Comments to the Author

Reviewer #1: In this manuscript, the authors applied high-pressure freezing in combination with transmission electron microscopy to study the ultrastructure of sperm development in subadult males of the praying mantid Hierodula membranacea. They revealed some features of cellular structures during spermatogenesis, including the structure of the chromosomes in prophase I and the distribution of organelles in spermatids. And they focused on the structure of the nebenkern in spermatids, revealing that the nebenkern of this species has two zipper-like membrane structures at opposing positions. However, there are some major issues with this paper.

1. The Abstract of this manuscript is too sample to attract readers. And the main point of this manuscript is not summarized in the Abstract.

2. Although the new methods were used to study the ultrastructure of sperm development, the topic of this manuscript was insufficient of innovation. Besides, the quality of all figures was not high, and several of them were blurry.

3. The function of the nebenkern during spermatogenesis was not mentioned in this manuscript, so the scientific value and research significance of this study was not outstanding.

4. Most of the refs cited in this paper are too old. More recent studies should be cited. And the format of the refs should be checked and revised carefully.

Reviewer #2: Peer review report on“Ultrastructure of the nebenkern during spermatogenesis in the praying mantid Hierodula membranacea”

This paper used state-of-the-art sample preparation techniques in combination with three-dimensional (3D) imaging such as high-pressure freezing and serial-section electron tomography, showing a survey of the steps of spermatogenesis in praying mantid Hierodula membranacea. The authors mainly focused on a structure of “nebenkern”, which was found in multiple insects during spermatogenesis. They observed the ultrastructure of nebenkern through three dimensions, in addition, they certified that the special organelle was composed of two interwoven segments that were connected by a zipper-like structure at opposing positions. This paper provided the detailed information of nebenkern for other in-depth study.

Comments to Author:

Minor comments:

1. In the manuscript of review versions, you don’t number the line in the whole paper. I hope you add the line number if you submit your manuscript in the future.

2. In the section of “Movie legends -Movie 4”, “Stitched” should be changed into “Stitched”.

3. In the section of “Figure legends”, there are many abbreviations of proper nouns, the full names appear just need appear once, you can use these abbreviations after the second.

4. It will be better to change your alpha code form lowercase to capital letter in your figures, because there are both capital letter to code your figures in “Figure legends” and manuscript.

5. Why you used the development stage of subadult male praying mantid as your object in this research, but not male praying mantid in other development stage? What advantages of the subadult mantid?

6. In your article, you focused on the “spermatids at different stage of development”, but I don’t know clearly the staging criteria. In addition, I think there is a little chaos in your “spermatocyte” and “spermatid”, you should distinguish the two stages more clearly.

7. Can you explain the differences of “axonemal microtubules” with “microtubules” for me?

8. I want to know more about the function of nebenkern during spermatogenesis in insects, I mean you should supply more information of nebenkern in your “Introduction” section. For example, its formation process, composition, function in reproduction and so on. This part is a little poor in your manuscript now.

9. In Figure 3, you mentioned that you get seven serial semi-thick (300 nm) sections, but there are just four sections you show?

10. Why you want to examine the microtubules in the 3D structure of nebenkern? In your opinion, if the cytoskeleton participates in the formation of nebenkern, I think it is necessary to examine the dynamical changes of microfilament during this process.

6. PLOS authors have the option to publish the peer review history of their article (what does this mean?). If published, this will include your full peer review and any attached files.

Reviewer #1: No

Reviewer #2: No

---

## [Author Response · Author response to Decision Letter 0]

8 Jun 2023

In the editorial response we were asked to remove the funding information from the acknowledgments section, which we did. In addition it was stated that the publication contains funding information that is not currently declared in the Funding Statement. This is not correct. All funding information was correctly entered and should not be changed or updated.

The repository for the raw data is now given as a link to be published along with the paper.

---

## [Decision Letter · Decision Letter 1]

4 Jul 2023

Ultrastructure of the nebenkern during spermatogenesis in the praying mantid *Hierodula membranacea*

PONE-D-23-11181R1

Dear Dr. Fabig,

We’re pleased to inform you that your manuscript has been judged scientifically suitable for publication and will be formally accepted for publication once it meets all outstanding technical requirements.

Kind regards,

Wan-Xi Yang, Ph.D.

Academic Editor

PLOS ONE

Additional Editor Comments (optional):

Reviewers' comments:

Reviewer's Responses to Questions

**Comments to the Author**

1. If the authors have adequately addressed your comments raised in a previous round of review and you feel that this manuscript is now acceptable for publication, you may indicate that here to bypass the “Comments to the Author” section, enter your conflict of interest statement in the “Confidential to Editor” section, and submit your "Accept" recommendation.

Reviewer #1: All comments have been addressed

Reviewer #2: All comments have been addressed

2. Is the manuscript technically sound, and do the data support the conclusions?

Reviewer #1: Yes

Reviewer #2: Yes

3. Has the statistical analysis been performed appropriately and rigorously? 

Reviewer #1: N/A

Reviewer #2: Yes

4. Have the authors made all data underlying the findings in their manuscript fully available?

Reviewer #1: Yes

Reviewer #2: Yes

5. Is the manuscript presented in an intelligible fashion and written in standard English?

Reviewer #1: Yes

Reviewer #2: Yes

6. Review Comments to the Author

Reviewer #1: (No Response)

Reviewer #2: (No Response)

7. PLOS authors have the option to publish the peer review history of their article (what does this mean?). If published, this will include your full peer review and any attached files.

Reviewer #1: No

Reviewer #2: No

---

## [Editor Report · Acceptance letter]

19 Jul 2023

PONE-D-23-11181R1 

Ultrastructure of the nebenkern during spermatogenesis in the praying mantid *Hierodula membranacea*

Dear Dr. Fabig:

I'm pleased to inform you that your manuscript has been deemed suitable for publication in PLOS ONE. Congratulations! Your manuscript is now with our production department. 

Kind regards, 

on behalf of

Dr. Wan-Xi Yang 

Academic Editor

PLOS ONE